# Green Synthesis of Al-ZnO Nanoparticles Using *Cucumis maderaspatanus* Plant Extracts: Analysis of Structural, Antioxidant, and Antibacterial Activities

**DOI:** 10.3390/nano14221851

**Published:** 2024-11-20

**Authors:** S. K. Johnsy Sugitha, R. Gladis Latha, Raja Venkatesan, Seong-Cheol Kim, Alexandre A. Vetcher, Mohammad Rashid Khan

**Affiliations:** 1Department of Chemistry, Holy Cross College, Nagercoil, Affiliated to Manonmaniam Sundaranar University, Tirunelveli 627012, TN, India; johnsysugitha@gmail.com; 2Department of Chemistry and Research Centre, Holycross College, Nagercoil 629002, TN, India; 3School of Chemical Engineering, Yeungnam University, Gyeongsan 38541, Republic of Korea; sckim07@ynu.ac.kr; 4Department of Biomaterials, Saveetha Dental College and Hospitals, SIMATS, Saveetha University, Chennai 600077, TN, India; 5Institute for Bionic Technologies and Engineering, I.M. Sechenov First Moscow State Medical University, Moscow 119991, Russia; avetcher@gmail.com; 6Institute of Pharmacy and Biotechnology (IPhB), Peoples’ Friendship University of Russia, Moscow 117198, Russia; 7Department of Pharmacology and Toxicology, College of Pharmacy, King Saud University, P.O. Box 2457, Riyadh 11451, Saudi Arabia; kmohammad@ksu.edu.sa

**Keywords:** green synthesis, doping, Al-ZnO nanoparticles, antioxidant, antibacterial activity

## Abstract

Nanoparticles derived from biological sources are currently garnering significant interest due to their diverse range of potential applications. The purpose of the study was to synthesize Al-doped nanoparticles of zinc oxide (ZnO) from leaf extracts of *Cucumis maderaspatanus* and assess their antioxidant and antimicrobial activity using some bacterial and fungal strains. These nanoparticles were analyzed using X-ray diffraction (XRD), ultraviolet–visible (UV-vis) spectroscopy, Fourier transform infrared spectroscopy (FTIR), scanning electron microscopy (SEM) with energy dispersive X-ray analysis (EDAX), transmission electron microscopy (TEM), and thermogravimetric analysis/differential thermal analysis (TG-DTA). The average crystalline size was determined to be 25 nm, as evidenced by the XRD analysis. In the UV-vis spectrum, the absorption band was observed around 351 nm. It was discovered that the Al-ZnO nanoparticles had a bandgap of 3.25 eV using the Tauc relation. Furthermore, by FTIR measurement, the presence of the OH group, C=C bending of the alkene group, and C=O stretching was confirmed. The SEM analysis revealed that the nanoparticles were distributed uniformly throughout the sample. The EDAX spectrum clearly confirmed the presence of Zn, Al, and O elements in the Al-ZnO nanoparticles. The TEM results also indicated that the green synthesized Al-ZnO nanoparticles displayed hexagonal shapes with an average size of 25 nm. The doping of aluminum may enhance the thermal stability of the ZnO by altering the crystal structure or phase composition. The observed changes in TG, DTA, and DTG curves reflect the impact of aluminum doping on the structural and thermal properties of ZnO nanoparticles. The antibacterial activity of the Al-ZnO nanoparticles using the agar diffusion method showed that the maximum zone of inhibition has been noticed against organisms of Gram-positive *S. aureus* compared with Gram-negative *E. coli*. Moreover, antifungal activity using the agar cup method showed that the maximum zone of inhibition was observed on *Aspergilus flavus*, followed by *Candida albicans*. Al-doping nanoparticles increases the number of charge carriers, which can enhance the generation of reactive oxygen species (ROS) under UV light exposure. These ROS are known to possess strong antimicrobial properties. Al-doping can improve the crystallinity of ZnO, resulting in a larger surface area that facilitates more interaction with microbial cells. The structural and biological characteristics of Al-ZnO nanoparticles might be responsible for the enhanced antibacterial activity exhibited in the antibacterial studies. Al-ZnO nanoparticles with *Cucumis maderaspatanus* leaf extract produced via the green synthesis methods have remarkable antioxidant activity by scavenging free radicals against DPPH radicals, according to these results.

## 1. Introduction

In recent years, biological approaches for the synthesis of nanoparticles have gained considerable interest because of their eco-friendly nature, simplicity, and cost-effectiveness [1]. They are feasible alternatives to conventional physical and chemical methods [2]. Among the different biological methods, plant extract-based synthesis of nanoparticles is emerging as popular, because it adheres to the 12 principles of green chemistry [3]. Further, it does not have to meet any specific conditions, as needed in other biological methods. Compared to chemical methods, biological methods are easiest [4,5]. This method is very fast, requires less energy, and gives better product yield [6]. These methods are environmentally friendly, economical, and safe, and they do not necessitate advanced equipment or chemicals. Biological synthesis can produce nanoparticles that are suitable for use in biomedicine and various other fields because of their biocompatibility [7].

*Cucumis maderaspatanus* is a part of Cucurbitaceae family, and one of its distinguishing features is the presence of cucurbitacins. These triterpenoid compounds are known for their potent biological activities, including anti-inflammatory, anticancer, and antimicrobial properties [8]. This plant is widely applied in folklore and traditional medicine. It is popularly known as Madras Thorn. Phytochemical analysis shows that the aqueous leaf extract of this plant contains alkaloids, proanthocyanidin, flavonoids, saponins, steroids, and phenolic groups, which act as reducing agents. These phytochemicals are responsible for the antioxidant, antimicrobial, and anticancer activities [9,10]. The plant also contains a variety of flavonoids and phenolic compounds, which are powerful antioxidants. The enhanced antioxidant and antibacterial properties suggest that these nanoparticles could be explored for drug delivery systems, where their biocompatibility and functional properties could improve therapeutic outcomes. The specific types and concentrations of these compounds can vary significantly from those found in other plants, potentially leading to different oxidative stress response mechanisms and enhanced antioxidant activity in the synthesized nanoparticles [11].

The presence of tannins in *Cucumis maderaspatanus* is another unique aspect. Tannins are polyphenolic compounds that can bind to proteins and other organic molecules, providing stabilization to the nanoparticles and contributing to their bioactivity. *Cucumis maderaspatanus* contain steroidal saponins, which are glycosides known for their ability to form stable complexes with metal ions. This property may enhance the formation and stability of Al-ZnO nanoparticles. The specific types of alkaloids found in *Cucumis maderaspatanus* can influence the biological activity of nanoparticles, including their antibacterial properties. This highlights their potential use in biomedical applications, such as wound healing, antimicrobial coatings, and the development of new antibacterial agents [12]. Terpenoids present in this plant play a role in reducing metal ions and stabilizing nanoparticles during synthesis. The presence of unique volatile oils and aromatic compounds in *Cucumis maderaspatanus* could impart distinctive properties to the synthesized nanoparticles, especially in terms of antimicrobial activity and potential applications in medicinal products [13].

The mineral content in *Cucumis maderaspatanus* might include trace elements that can contribute to the doping process or act as co-factors in the reduction and stabilization of nanoparticles, offering advantages over other plants. The specific combination and interaction of the phytochemicals in *Cucumis maderaspatanus* may be unique, leading to enhanced or novel properties in the synthesized nanoparticles. This synergistic effect might not be present in other plant extracts, making the use of this plant particularly beneficial. These substances can facilitate the reduction of metal ions to their respective metal nanoparticles. In the case of ZnO and Al-ZnO synthesis, the plant extract helps to reduce zinc and aluminum ions to form nanoparticles. The compounds in *Cucumis maderaspatanus* plant extracts can also act as stabilizers, preventing the agglomeration or clumping of nanoparticles. This helps in maintaining the stability and uniformity of the synthesized nanoparticles.

In this article, we discuss the biological method for the green synthesis of Al-ZnO nanoparticles viz. *Cucumis maderaspatanus* plant extract. XRD, UV-vis, FTIR, SEM, EDAX, TEM, and TGA/DSC methods were among the characterization methods used to assess the produced Al-ZnO nanoparticles. Employing the agar well diffusion method, the antimicrobial properties were established. The antioxidant activities of the plant extract and nanoparticles were analyzed by using the 2,2-diphenyl-1-picrylhydrazyl (DPPH) assay. The novelty of the current study is the investigation of the biological activities of synthesized Al-ZnO nanoparticles utilizing *Cucumis maderaspatanus* leaf extracts. Zinc acetate dihydrate and aluminum acetate were used as the precursors, and the leaf extract acted as the reducing agent. The nature of nanoparticles were explained by characterizations like XRD, UV-vis spectroscopy, FTIR, SEM with EDAX, TEM, and TGA/DSC techniques. Additionally, to improve the therapeutic activity, an antimicrobial and antioxidant study was also performed.

## 2. Materials and Methods

### 2.1. Materials

Fresh *Cucumis maderaspatanus* leaves were gathered in Kanyakumari, Tamil Nadu, India. All the materials utilized in the studies were of analytical grade and were used as such without further refinement. Zn(CH_3_COO)_2_·2H_2_O, aluminum acetate, and sodium hydroxide (98%) were received from Merck Chemicals, Mumbai, India.

### 2.2. Preparation of Cucumis maderaspatanus Leaf Extracts

The plant leaves were properly cleaned with distilled water to get rid of the dust particles. The leaves were washed and then left to dry in the shade. After that, an electric blender was used to turn it into powder. Ten g of the powder was mixed with 10 milliliters of de-ionized water and heated for 30 min at 70 °C. After that, Whatman filter paper was used to filter it. Then, the filtrate was collected in an amber bottle and kept at 4 °C.

### 2.3. Green Synthesis of Al-ZnO Nanoparticles Using Cucumis maderaspatanus

The green synthesis of Al-ZnO nanoparticles from a solution of *Cucumis maderaspatanus* leaf extract is shown in Figure 1. In 20 milliliters of distilled water, 2 g of zinc acetate dihydrate, 2 g of aluminum acetate, and 2 g of NaOH were combined. Ten milliliters of plant extract was then added and stirred continuously for two hours at room temperature. By adding NaOH solution (1.0 N) dropwise, the pH of the mixture was brought to 12. A light-yellow solution was produced after 3 h of constant stirring, and it was left undisturbed for a day. A pale-yellow precipitate then developed. In addition to ethanol, distilled water can be used repeatedly to eliminate any residual pollutants or organic materials. This solution was then fully dried and calcined at 250 °C for 2 h [14]. By using calcinations, the contaminants in the precipitate were eliminated. Subsequently, the specimen was ground into a fine powder. Then, the sample was used for different characterization techniques such as XRD, UV-vis, FTIR, SEM with EDAX, TEM, TG, and DTA/DTG and also investigated with biological studies such as antibacterial, antifungal, and antioxidant.

### 2.4. Characterization of Al-ZnO Nanoparticles

#### 2.4.1. UV-Vis Spectroscopy Analysis

The Shimadzu UV-2450 spectrophotometer (Kyoto, Japan), was used for the UV-vis spectral studies. A quartz cuvette of 1 cm in path length was used for the analysis. The formation of nanoparticles during various synthetic processes was recorded in the range of 200–800 nm.

#### 2.4.2. FOURIER Transform Infrared Spectroscopy (FTIR) Analysis

For FTIR analysis, Perkin Elmer-Spectrum Two spectrometers (Waltham, MA, USA) were used in the range of 500–4000 cm^−1^. The dried samples were used for FTIR analysis.

#### 2.4.3. X-Ray Diffraction (XRD) Analysis

For X-ray diffraction pattern measurements, the samples were prepared by drop-casting a few drops of nanoparticle solution in a glass slide and drying. The measurements were carried out on a Bruker AXS D8 Advance Diffractometer (Billerica, MA, USA), with Cu radiation (λ = 1.5406 Å).

#### 2.4.4. Scanning Electron Microscopy (SEM) Analysis

SEM analysis was carried out on a JEOL JEM-2100 (Tokyo, Japan) microscope. For the analysis, the samples were prepared by drying a few drops of aqueous nanoparticle solution on a carbon-coated copper grid and allowing them to dry at room temperature.

#### 2.4.5. Transmission Electron Microscopy (TEM) Analysis

The morphology and particle size, shape, and distribution of Al-ZnO NPs in powder form were analyzed using high-resolution transmission electron microscopy (HRTEM) on a JEM-2100 instrument manufactured by JEOL in Tokyo, Japan, operating at an accelerated voltage of 200 kV.

#### 2.4.6. Dynamic Light Scattering, and Zeta Potential Analysis

The dynamic light scattering (DLS) and zeta potential analysis were used to confirm both the size and surface charge of Al-ZnO nanoparticles that were synthesized (Malvern Instruments, Malvern, UK).

#### 2.4.7. Thermogravimetric Analysis (TGA)

The thermogravimetric–differential thermal analysis was carried out to assess the thermal properties (TA Instruments, SDT Q6000, New Castle, DE, USA). In a nitrogen air, the TG-DTA was conducted from 50 °C to 800 °C at a scanning rate of 20 °C/min (nitrogen flow rate was 60 mL min^−1^).

#### 2.4.8. Antibacterial Activity Studies

The antibacterial activity of Al-ZnO nanoparticles was assessed against three Gram-negative bacterial species: *Klebsiella pneumoniae*, *Escherichia coli*, and *Vibrio cholerae*, as well as three Gram-positive organisms: *Staphylococcus aureus*, *Bacillus subtilis*, and *Streptococcus mutans*. The bacterial strains were maintained in Brain Heart Infusion (BHI) broth at −20 °C. A 300 mL of each stock culture was added to 3 mL of BHI broth. Overnight cultures were incubated at 36 °C ± 1 °C for 24 h, and the purity of the cultures was checked after 8 h of incubation [15]. After 24 h, the bacterial suspensions (inoculum) were diluted with sterile physiological solution to a final concentration of 10^8^ CFU/mL (turbidity equivalent to the McFarland barium sulfate standard 0.5) for diffusion and indirect bioautographic tests. For the direct bioautographic test, the bacterial suspension was diluted with BHI broth to 10^9^ CFU/mL (McFarland standard). An indicator solution for bacterial growth determination was prepared by dissolving 2 mg/mL of 2-(4-iodophenyl)-3-(4-nitrophenyl)-5-phenyltetrazolium chloride in a 70% ethanol solution. This solution was used in the bacterial growth tests.

#### 2.4.9. Antioxidant Activity Studies

The radical scavenging activity (RSA) of the sample was performed by the 2,2-diphenyl-1-picrylhydrazyl (DPPH) method. In brief, 3.5 mL of 0.1 mM DPPH containing different concentrations of sample powder were subjected to sonication before incubation at 37 ± 2 °C for 30 min under dark conditions. The absorbance of the incubated sample was read at 517 nm in a spectrophotometer, and percent RSA was determined.

## 3. Results and Discussion

### 3.1. Structural and Morphological Studies

Figure 2A, represents the XRD spectrum of Al-ZnO nanoparticles utilizing *Cucumis maderaspatanus* leaf extracts. The noticeable broadening of the diffraction lines clearly indicates that the synthesized powders are at the nanoscale. In Figure 2A, the diffraction peaks appear at 31.77 °C, 34.40 °C, 36.24 °C, 47.49 °C, 56.72 °C, 62.77 °C, 66.29 °C, 67.86 °C, 69.01 °C, 72.56 °C, 76.85 °C, 81.259 °C, 89.470 °C, 92.63 °C, 95.147 °C, and 93.487 °C. These peaks align closely with the hexagonal wurtzite structure typical of Al-ZnO nanoparticles (JCPDF 36–1451). The average crystallite size of the synthesized particles was determined using the Scherrer equation [16].
D = k λ/βCos θ(1)

In this equation, k is the shape factor with a value of 0.94, λ represents the wavelength of the X-ray source, β is the full width at half maximum (FWHM) of the diffraction peak, and θ is the Bragg angle corresponding to the peak with the highest intensity in the XRD pattern. The average crystallite size was calculated to be approximately 25 nm. Similar findings have been reported for other samples of Al-ZnO nanoparticles [17]. Figure 2B presents scanning electron microscopy (SEM) images of the Al-ZnO nanoparticles, which show uniform distribution according to the SEM analysis. Comparable results have been observed in other Al-ZnO nanoparticles samples [18]. The composition of the Al-ZnO nanoparticles was analyzed using energy dispersive X-ray spectroscopy (EDAX). Figure 2C displays the EDAX spectrum, confirming the presence of Zn, Al, and O within the ZnO structure. However, the concentration of Al is relatively lower compared to Zn and O. This lower concentration is likely due to the minimal infusing level of Al in comparison to the primary ZnO nanoparticles. The data suggest that a significant proportion of Al atoms were not incorporated into the ZnO nanostructure [19] due to several factors, including the solubility limits of Al in ZnO, suboptimal synthesis conditions, size and structural effects, limitations of the characterization technique, and potential chemical interactions [20]. Ensuring optimal doping conditions and through characterization can help address these issues and improve the efficiency of dopant incorporation [21]. This situation is compatible with XRD results. If XRD patterns show no significant changes or additional phases, or if the lattice parameters remain unchanged, it aligns with the observation that Al is not effectively incorporated into the ZnO nanostructure. The SEM image of Al-ZnO nanoparticles obtained from *Cucumis maderaspatanus* leaf extracts reveals the uniform distribution of nanoparticles synthesized, and it confirms the particle size.

The formation of biosynthesized Al-ZnO nanoparticles was confirmed through high-resolution TEM analysis, revealed in Figure 3A,B. ZnO naturally forms a hexagonal wurtzite crystal structure, so this observation suggests that the nanoparticles have retained the typical ZnO crystal structure even after infusion with aluminum [22]. The results indicate that the green synthesized Al-ZnO nanoparticles exhibit hexagonal shapes with an average particle size of 25 nm. For Al-ZnO, HR-TEM might show slight distortions or defects due to the incorporation of Al, but if the hexagonal structure is well-defined, it indicates that the overall ZnO structure is preserved [23]. This characterization supports the effectiveness of the biosynthesis method and provides valuable information about the nanoparticle’s morphology and structure [24]. Additionally, the lattice fringes are clearly visible without distortion, suggesting that the aluminum-infused zinc oxide nanoparticles possess high crystallinity. The selected area electron diffraction SAED pattern (Figure 3C) displays a set of concentric rings with bright spots, suggesting the highly crystalline nature of the Al-ZnO nanoparticles [25,26]. In Figure 3C, Al-ZnO nanoparticles’ hexagonal wurtzite crystalline structure is further confirmed by the peaks in the XRD pattern and the diffraction rings observed in the SAED image. The clear visibility of lattice fringes in the TEM images indicates that the Al-ZnO nanoparticles possesses high crystallinity [27,28]. The TEM image of Al-ZnO nanoparticles derived from *Cucumis maderaspatanus* leaf extracts is presented in Figure 3, and it also confirms the particle size and shows a uniform distribution of the produced nanoparticles.

### 3.2. FTIR Analysis

The functional groups present in the synthesized Al-ZnO nanoparticles were identified using Fourier transform infrared (FTIR) spectroscopy. The FTIR spectrum was recorded with a PerkinElmer spectrometer, operating at a resolution of 4 cm^−1^ across the 4000–400 cm^−1^ range. Figure 4 displays the FTIR spectrum for the Al-ZnO nanoparticles. A broad band at 3468 cm^−1^ is attributed to O-H bonding. The peak observed at 2058 cm^−1^ is associated with the C=O group in anhydrides, and the peak at 1744 cm^−1^ indicates the presence of a C=O group from ketones. The peak at 1712 cm^−1^ suggests the presence of an amide C=O group. while the peak at 1417 cm^−1^ is indicative of CH_3_ bending vibrations.

The peak at 982 cm^−1^ is attributed to C=C bending vibrations in alkenes. Similar findings have been reported in other samples of doped ZnO nanoparticles. FTIR analysis in this investigation indicates that phenolic compounds in flavonoids exhibit a stronger affinity for metals. This implies that the phenolic group may assist in the formation of metal nanoparticles while stabilizing the solution by preventing the aggregation of particles. The surface contact of groups such as O-H, C=O, and C=C bending and stretching suggests the use of this nanoparticle in photocatalysis, sensors, and catalysis [29].

### 3.3. Optical Studies

The UV–visible spectrum of the Al-ZnO is shown in Figure 5A. In many experiments, particularly preliminary studies or exploratory research, the measurement setup may not be standardized. This means that the absorbance readings can vary based on the specific equipment, calibration, or sample conditions used, making absolute quantification less meaningful. The presence of other components in the sample that absorb light at similar wavelengths can contribute to a background signal, complicating the interpretation of specific absorbance. The sample absorbs radiation in the UV range up to 351 nm, and almost all the visible-spectrum radiation is transmitted by the ZnO nanoparticles. The fact that the sample absorbs radiation up to 381 nm suggests that the Al-ZnO nanoparticles have an optical absorption edge in the UV range. Al-ZnO nanoparticles typically exhibit strong UV absorption due to their wide bandgap, which usually ranges from about 3.2 to 4.0 eV [30]. This bandgap allows ZnO to absorb UV light efficiently. The bandgap was determined using Tauc’s plot. For the Al-ZnO nanoparticles, the bandgap was calculated by extrapolating the curve from the plot of (hν) versus (αhν)^2^, as illustrated in Figure 5B. In this case, ν denotes the frequency, and α represents the optical absorption coefficient. The extrapolated bandgap energy was found to be approximately 3.25 eV, which is slightly lower than the 3.3 eV typically reported for undoped ZnO. This reduction is likely due to the effects of vacancies and dopant atoms. As an n-type semiconductor, ZnO shows a decrease in bandgap due to electron transitions from the valence band to the conduction band [31].

Al-ZnO nanoparticles are often used to modify electronic and optical properties. While the primary bandgap of ZnO is typically in the UV range, doping can sometimes shift the absorption edge slightly. However, if the absorption edge remains close to 381 nm, it suggests that the infusion has not significantly altered the fundamental bandgap of ZnO. The absorption up to 381 nm confirms that the Al-ZnO nanoparticles have a significant UV absorption edge, consistent with a bandgap of around 3.25 eV. This suggests that the doping does not drastically alter the intrinsic bandgap of ZnO [32]. Overall, the optical properties of Al-ZnO nanoparticles, as indicated by UV absorption and visible light transmission, align with the expected behavior of ZnO, showing that infusion has not severely disrupted the fundamental optical characteristics. These properties highlight their suitability for applications requiring UV absorption combined with visible light transparency [33]. The infusing of aluminum with zinc oxide not only maintains a bandgap close to that of pure ZnO but also enhances its photocatalytic and antimicrobial properties, making it more effective for various applications compared to undoped ZnO. The bandgap energy allows for effective excitation of electrons, facilitating redox reactions. ZnO nanoparticles with a bandgap of 3.25 eV and an absorption band at 381 nm have significant potential in various field, including photocatalysis, UV protection, sensors, optoelectronics devices such as light-emitting diodes (LEDs) and laser diodes, particularly in the UV range, and antimicrobial applications [34].

### 3.4. Dynamic Light Scattering and Zeta-Potential Analysis

The most effective method for measuring the particle size is the dynamic light scattering method (DLS). In regard to volume and intensity, the Al-ZnO nanoparticles which are produced are evenly distributed. The surface charge of biologically synthesized Al-ZnO nanoparticles can be determined via zeta-potential analysis. The strength of the charge has an association with the stability of the nanoparticle. The 63–105 nm particle size of the synthesized nanoparticles, as seen in Figure 6A, indicates its excellent stability. The zeta-potential value, if positive or negative, indicates enhanced physical colloidal stability. The stability of Al-ZnO nanoparticles and their efficient electric charge on the surface were measured via zeta-potential (Figure 6B). The zeta-potentials that are shown are 33.16 mV. As a result, the synthetic Al-ZnO nanoparticles’ zeta-potential value is almost physically stable.

### 3.5. Thermal Analysis

Thermogravimetric analysis/differential thermal analysis (TG/DTA) was used to confirm the formation and evaluate the thermal stability of the Al-ZnO nanoparticles that were produced using leaf extracts from *Cucumis maderaspatanus*. The TG/DTA curve of Al-ZnO NPs is shown in Figure 7 and was obtained between 50 and 800 °C in an air atmosphere at a heating rate of 10 °C/min.

Al-ZnO nanoparticles’ thermal degradation using leaf extracts from *Cucumis maderaspatanus* happened in multiple stages, and following thermal analysis up to 800 °C, 6.41% residue was left behind. The thermal dehydration of Al-ZnO NPs was the cause of the first weight loss, which was 18.17% from room temperature to 168.6 °C. Water molecules were lost during this thermal dehydration process. The sample showed a second weight loss of 14.19 percent after being heated to 262.4 °C. The breakdown of some of the organic moieties in the phytochemicals acting as capping agents and reductants was responsible for the sample’s second weight loss of 14.19 percent when it was heated to 262.4 °C. Around 473 °C, a final weight loss of 61.23% was noted; however, no more weight loss was noted when the temperature was raised to 800 °C. Significant weight loss events in the TGA curve correspond to the blue curve, which depicts DTG and shows weight change rates with peaks at 300 °C and 450 °C. Critical temperature points for abrupt changes in weight loss are indicated by the DTG peaks.

### 3.6. Biological Studies of Al-ZnO Nanoparticles Utilizing Cucumis maderaspatanus Leaf Extracts

#### 3.6.1. Antibacterial Activity

The inhibitory effect of the sample of *Cucumis maderaspatanus* from species *Klebsiella bacillus* and *Bacterium coli*, *V. cholereae*, *S. aureus*, *B. subtilis*, and *S. mutans* at a concentration of 10 mg/mL results in different extents of inhibition. Apart from the study, the maximum inhibitory effect of antibacterial activity in the resistant zone (2.65 ± 0.15) has been noticed against *Klebsiella bacillus* at a concentration of 10 mg/mL. Significantly, a *Cucumis maderaspatanus* sample shows that maximum antibacterial activity has been observed against the Gram-negative organism *Klebsiella bacillus* compared to other tested organisms [35], and the values are presented in Table 1. This suggests that the extract contains bioactive compounds that are particularly effective against this pathogen.

*Cucumis maderaspatanus* may contain various phytochemicals, such as flavonoids, saponins, terpenoids, or alkaloids, that exhibit antibacterial properties. The fact that this activity is reported as being higher than against other tested organisms implies that *Klebsiella bacillus* is more susceptible to the compounds in the *Cucumis maderaspatanus* extract compared to other bacterial strains tested. This specificity is important for understanding the potential therapeutic applications of the extract. *Cucumis maderaspatanus* may contain various phytochemicals, such as flavonoids, saponins, terpenoids, or alkaloids, that exhibit antibacterial properties [36]. The efficacy of these compounds can vary depending on their ability to interact with bacterial cell components. The observation that *Cucumis maderaspatanus* extract exhibits maximum antibacterial activity against *Klebsiella bacillus* compared to other tested organisms suggests that it has potent antibacterial properties, particularly effective against Gram-negative bacteria. This finding underscores the potential of *Cucumis maderaspatanus* as a source of natural antibacterial agents, with implications for treatment and prevention strategies, particularly for diseases like cholera. Further research will be necessary to explore the specific bioactive compounds responsible for this activity and their mechanisms of action [37]. Interestingly, it is expressed as the potential value of inhibitory activity against the same organism as the positive control of antibiotic disc gentamycin.

The antibacterial properties of Al-ZnO nanoparticles have been widely studied because of their numerous potential applications. As reported by Rekha et al., infusing ZnO nanoparticles with manganese enhanced their antibacterial effectiveness against both Gram-positive and Gram-negative bacteria [38]. It was observed that the crystalline size is inversely proportional to susceptibility to bacterial action and is directly related to silver concentration [39]. A separate study by Yang et al. found that the antibacterial activity of green-synthesized ZnO increases in direct proportion to the concentration of nanoparticles [40]. The figure shows the zones of inhibition for *S. aureus* at concentrations of 5 mg/mL and 10 mg/mL.

This study examines the antibacterial properties of Al-ZnO nanoparticles, synthesized at 250 °C for 2 h. Antibacterial activity was studied with bacteria such as *Staphylococcus aureus* (Gram-positive) and *Escherichia coli* (Gram-negative), as well as *Aspergillus flavus*, and *Candida albicans*. The resistant zone of Al-ZnO nanoparticles is shown in Figure 8. Green synthesis often results in smaller, more uniform particles due to the controlled conditions provided by plant extracts. The shape of nanoparticles can be influenced by various factors, including concentration of precursor materials; i.e., higher concentrations can lead to different growth rates, affecting shape. pH and temperature parameters can dictate the nucleation and growth processes, influencing morphology. Different extracts from *Cucumis maderaspatamus* may contain various phytochemicals that can affect particle shape. Al-ZnO nanoparticles have a higher surface area, and previous studies have shown that an increased surface area of nanoparticles enhances their antibacterial activity. Hexagonal shapes can enhance penetration into bacterial membranes, improving antibacterial efficacy, and also provide a larger surface area and increased interaction with bacteria [41,42,43]. The bactericidal effect is greatly influenced by the size and shape of the particles. Al-ZnO nanoparticles showed increased effectiveness against *S. aureus* (Gram-positive) bacteria compared to *V. cholerae* by producing toxic oxygen radicals and penetrating the cell wall, thereby inhibiting bacterial growth. The negatively charged bacteria attached to the positively charged aluminum and zinc within the intracellular material, leading to bacterial death. The antibacterial effect at a concentration of 5 mg/mL suggests that even low concentrations of this nanoparticle can be effective, which can be beneficial in reducing costs and potential toxicity in practical applications. According to the current study, *Cucumis maderaspatanus* plant extract was utilized to synthesize Al-ZnO nanoparticles, with antibacterial characteristics. In addition, we will demonstrate that further biological study is required to analyze the effects of plant extract and aluminum doping on antibacterial property.

#### 3.6.2. Antifungal Activity

The antifungal activity of aluminum-infused zinc oxide nanoparticles is a topic of interest in materials science and applied microbiology. This study shows that the standard drug amphotericin B has the highest antifungal activity for *Aspergillus flavus* (1.2 ± 0.01) in 10 mg/mL, followed by 0.73 ± 0.04 in 5 mg/mL. Compared with the standard drug, no effect occurs with the fungus *C. albicans*. This result clearly shows that whenever the concentration of the sample increases, the antifungal activity also noticeably increases [44]. The antifungal results are tabulated in Table 2. No antifungal effect is observed with amphotericin B for *Candida albicans*, meaning the drug does not inhibit the growth of this fungus under the tested conditions. *C. albicans* could be resistant to amphotericin B in this particular strain or under the specific conditions tested. There might be experimental factors affecting the results, such as the method of application, medium, or the specific strain of *C. albicans* used [45]. If these nanoparticles demonstrate significant antifungal activity against *Aspergillus flavus* compared to existing treatments or controls, this could indicate that the nanoparticles are an effective agent against this particular fungal species. However, *Candida albicans* showed no response to this sample.

The data shows a decrease in the inhibition or effect size from 10 mg/mL to 5 mg/mL (1.2 to 0.73). This concentration-dependent effect could be novel if it suggests that the effectiveness of the nanoparticles can be fine-tuned by adjusting their concentration, allowing for optimization in real-world applications. If the nanoparticles provide the desired antifungal effects at lower concentrations than traditional antifungal agents, they might be useful in reducing potential toxicity and side effects, representing a novel approach to treatment. The novelty might also lie in the composition or synthesis method of the nanoparticles. If they are made from new materials or through novel methods that enhance their antifungal properties, this would contribute to the scientific and medical fields. If the nanoparticles are shown to have a synergistic effect when combined with other antifungal agents, this could represent a novel finding that improves the overall antifungal strategy against Aspergillus flavus. If the nanoparticles are designed for targeted delivery of antifungal agents directly to the site of infection, minimizing systemic exposure and potential side effects, this targeted approach could be considered novel.

#### 3.6.3. Antioxidant Activity

The aluminum-infused zinc oxide nanoparticles’ antioxidant activity is determined through a radical scavenging method utilizing DDPH as the free radical. This approach relies on antioxidants scavenging DPPH, leading to the decolonization of the DPPH methanol solution through a reduction reaction. In this study, the findings demonstrate a color change in the DPPH methanol solution from deep violet to either colorless or pale yellow. Figure 9 depicts the variation of the chart corresponding to the percentage of inhibition vs. concentration. From the chart, it is clear that as the concentration of aluminum-infused zinc oxide nanoparticles increases, there is a corresponding rise in radical scavenging activity. A maximum radical scavenging activity (58.32%) is observed at a concentration of 1.0 mg/mL. Table 3 indicates the antioxidant result of aluminum-infused zinc oxide nanoparticles. Previous reports have indicated that enhancing antioxidant activity is achievable through the incorporation of Al into the ZnO lattice, leading to the elimination of free radicals through a synergistic effect. Functional groups like phenolic and flavonoids found in the aluminum-infused zinc oxide nanoparticles synthesized from leaf extract contribute to boosting its antioxidant capacity [46]. Infusing aluminum with zinc oxide introduces extra charge carriers (electrons) into the ZnO nanoparticles. This increase in charge carrier density can enhance the generation of electron–hole pairs when the material is exposed to light. These charge carriers can participate in redox reactions, leading to an increase in the formation of reactive oxygen species (ROS) such as hydroxyl radicals (•OH) and superoxide anions (•O_2_^−^), which are involved in radical scavenging [47]. Infusion can modify the surface properties of ZnO, including increasing its surface area and improving its surface reactivity. This can lead to more active sites available for radical scavenging reactions. The introduction of Al into ZnO can alter the band structure and energy levels of the material. This change can affect the generation and behavior of reactive species, which can enhance the material’s ability to scavenge radicals [48].

Aluminum-infused zinc oxide nanoparticles are a relatively novel compound compared to conventional antioxidants. ZnO nanoparticles have been studied for various applications, but infusing them with aluminum (Al) could introduce new or enhanced properties, such as improved radical scavenging activity due to changes in electronic structure, surface chemistry, or particle size [49,50,51]. The observed radical scavenging activity (61.05%) at a specific concentration suggests that aluminum-infused zinc oxide nanoparticles could be effective in neutralizing free radicals, which are known to cause oxidative stress and damage in biological systems. This level of activity might be higher or more efficient than other materials at similar concentrations, indicating potential superiority as an antioxidant. Preliminary studies indicated that increasing concentrations beyond 1.0 mg/mL could lead to diminishing returns in activity, possibly due to saturation effects where the active compounds may not effectively interact with the target microorganisms. The identification of a maximum activity at 1.0 mg/mL shows a precise understanding of the concentration at which these nanoparticles are most effective [52]. This knowledge is valuable for designing applications where optimal antioxidant effects are desired, such as in the biomedical, cosmetic, or food industries.

The concentration of 1.0 mg/mL demonstrates the best antioxidant activity among the tested concentrations, because it was observed that antioxidant activity peaks at that concentration, and further increases might not provide additional benefit or could intro-duce complications (like toxicity, aggregation, or solubility issues). Given the antioxidant properties, aluminum-infused zinc oxide nanoparticles could have applications in fields like medicine (for therapeutic uses against oxidative stress-related conditions), food preservation (to enhance shelf life by preventing oxidation), or materials science (to create coatings or materials that resist degradation). Infusing aluminum into zinc oxide could alter increased stability, biocompatibility, or specific interactions with radicals.

## 4. Conclusions

Al-ZnO nanoparticles were synthesized via a green synthesis method using leaf extract from *Cucumis maderaspatamus*. The characterization of the resulting nanoparticles was conducted using several techniques, including UV-vis spectroscopy, FTIR, XRD, SEM with EDAX, TEM, TG, DTA, and DTG analysis. XRD confirmed the hexagonal wurtzite structure of the Al-doped zinc oxide nanoparticles, while SEM images demonstrated a uniform distribution with an average crystal size of 25 nm. The EDAX analysis indicated the appropriate ratios of Zn, Al, and O. TEM further corroborated the average crystalline size of 25 nm. The UV-vis spectrum showed an absorption peak around 381 nm, and the Tauc relation indicated a bandgap of 3.25 eV. FTIR analysis confirmed the presence of hydroxyl groups, C-O stretching from methylene groups, and C=C stretching from alkene compounds. The incorporation of aluminum is suggested to enhance the thermal stability of ZnO nanoparticles by potentially modifying their crystal structure or phase composition. TG, DTA, and DTG studies providing valuable insights into the way aluminum doping influences the thermal properties, revealing variations in stability and possible novel phases or reactions. The current work demonstrates the antimicrobial activity of Al-ZnO nanoparticles synthesized using *Cucumis maderaspatanus* plant extract. We will also clarify that further comparative biological tests are required to assess the individual contributions of the plant extract and aluminum doping on antimicrobial activity. This addition will ensure transparency regarding the scope and limitations of the current study. The antibacterial activity using the agar diffusion method demonstrated a significant zone of inhibition against Gram-positive *Staphylococcus aureus*, outperforming that observed for Gram-negative bacteria. In antifungal testing via the agar cup method, *Aspergillus flavus* exhibited a greater resistant zone (1.25 ± 0.01) compared to *Candida albicans*. These findings suggest that aluminum incorporation enhances antimicrobial efficacy, likely due to the phytochemicals present in the extract. In terms of antioxidant activity, a peak radical scavenging rate of 58.32% was noted at a concentration of 1.0 mg/mL, which can be attributed to reactive groups such as phenolic and flavonoid compounds found in the aluminum-infused zinc oxide nanoparticles derived from the extract. Overall, these results suggest potential applications of these nanoparticles in personalized medicine, targeted drug delivery systems, and innovative therapeutic strategies.

## Figures and Tables

**Figure 1 nanomaterials-14-01851-f001:**
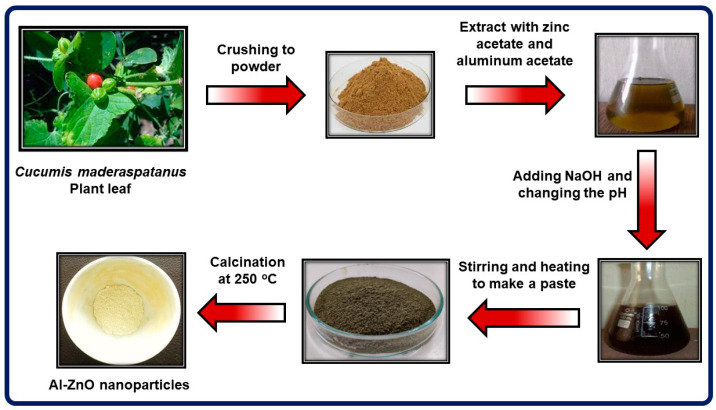
Schematic diagram of Al-ZnO nanoparticles using *Cucumis maderaspatanus* leaf extract.

**Figure 2 nanomaterials-14-01851-f002:**
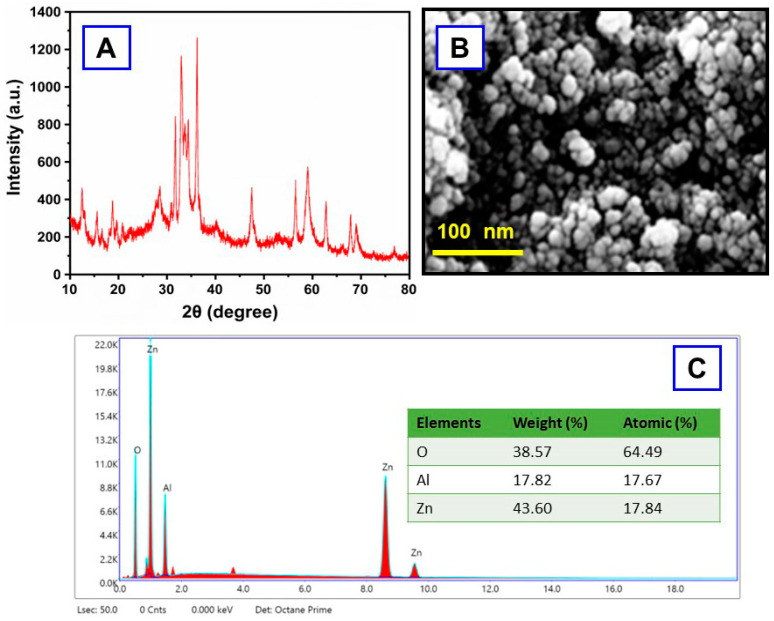
Al-ZnO nanoparticles derived from *Cucumis maderaspatanus* leaf extracts: (**A**) XRD pattern, (**B**) SEM, (**C**) EDAX.

**Figure 3 nanomaterials-14-01851-f003:**
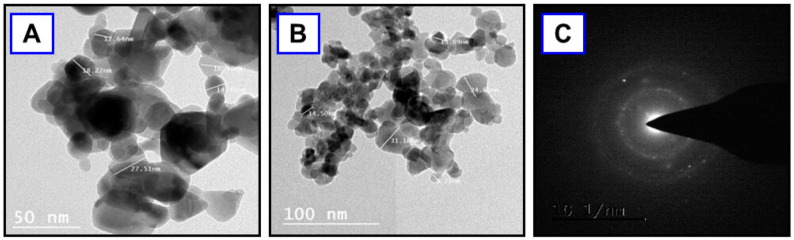
(**A**) and (**B**) TEM images of Al-ZnO nanoparticles, (**C**) SAED pattern of Al-ZnO nanoparticles using *Cucumis maderaspatanus* leaf extracts.

**Figure 4 nanomaterials-14-01851-f004:**
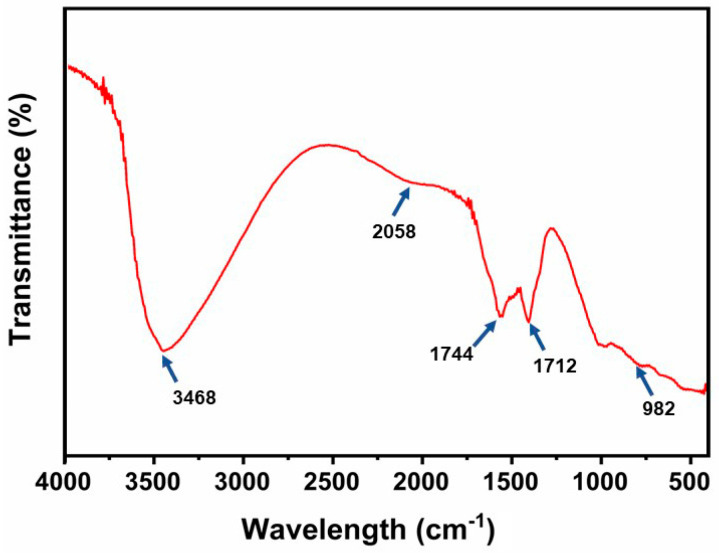
FTIR spectrum of the Al-ZnO nanoparticles using *Cucumis maderaspatanus* leaf extracts.

**Figure 5 nanomaterials-14-01851-f005:**
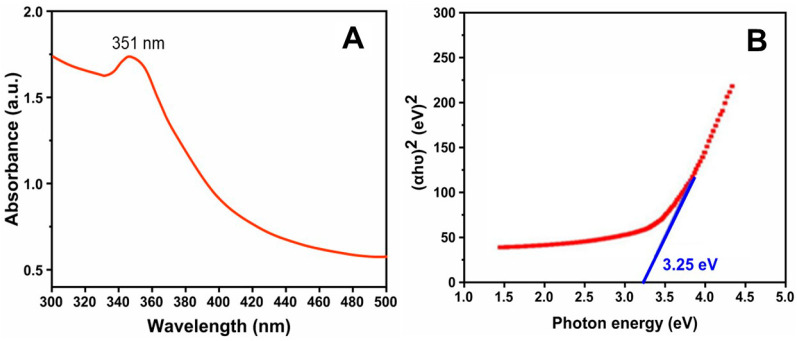
(**A**) UV spectrum, (**B**) Tauc plot of Al-ZnO nanoparticles using *Cucumis maderaspatanus* leaf extracts.

**Figure 6 nanomaterials-14-01851-f006:**
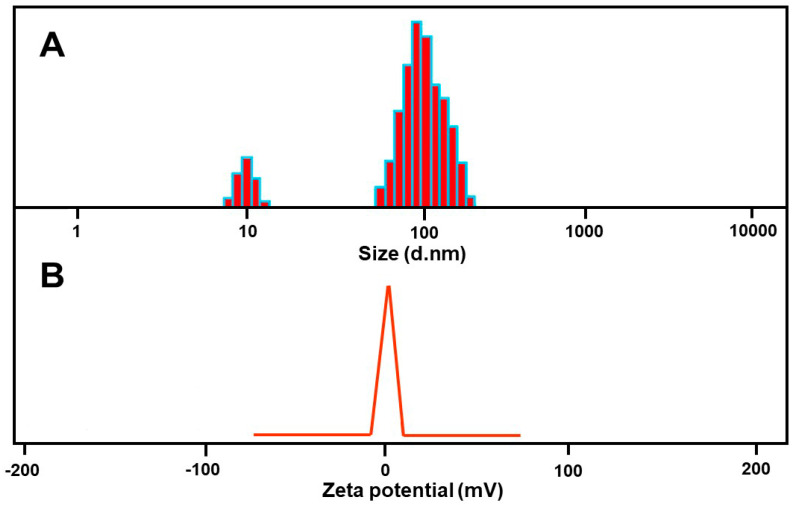
(**A**) Dynamic light scattering (DLS), and (**B**) zeta-potential measurement of Al-ZnO nanoparticles using *Cucumis maderaspatanus* leaf extracts.

**Figure 7 nanomaterials-14-01851-f007:**
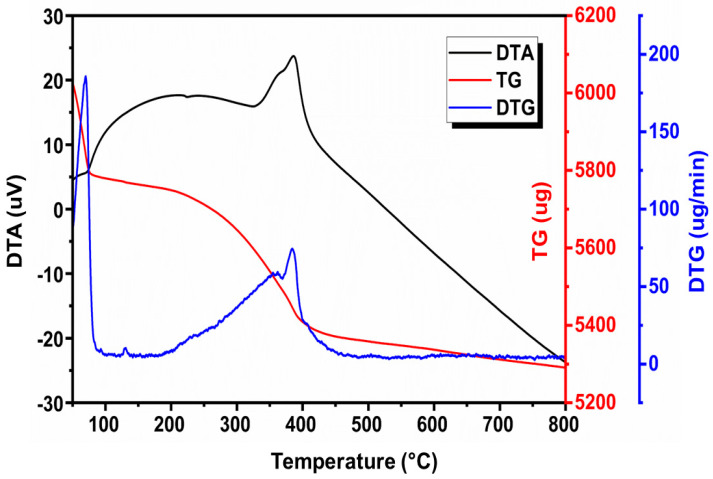
TG/DTA curves of Al-ZnO nanoparticles using *Cucumis maderaspatanus* leaf extracts.

**Figure 8 nanomaterials-14-01851-f008:**
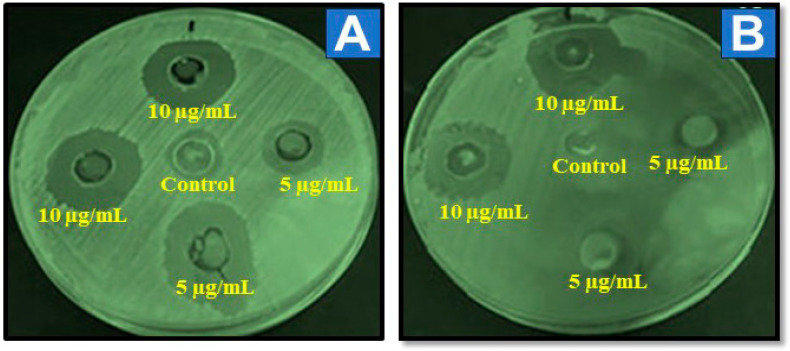
*Cucumis maderaspatanus* leaf extracts served to synthesize Al-ZnO, which exhibited antibacterial activity against (**A**) *S. aureus* and (**B**) *B. subtilitis*.

**Figure 9 nanomaterials-14-01851-f009:**
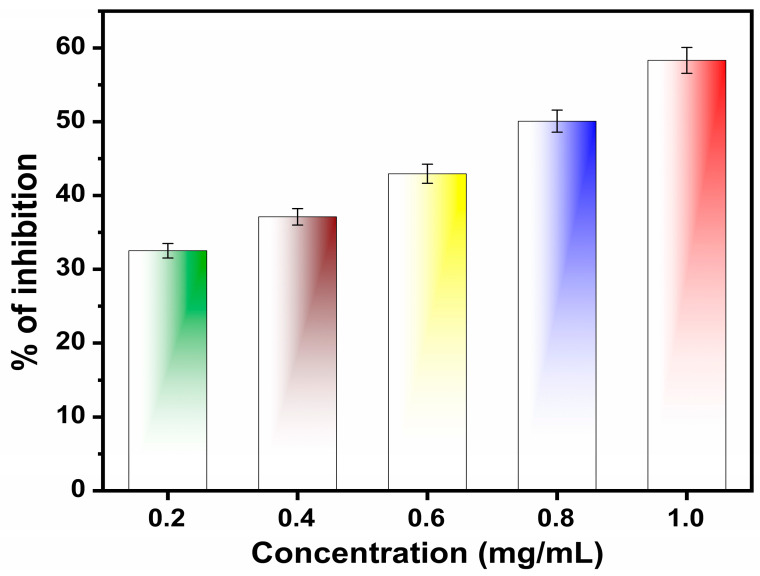
The concentration verses antioxidant activity of % of inhibition Al-ZnO nanoparticles using the DPHH free radical assay method.

**Table 1 nanomaterials-14-01851-t001:** Antimicrobial results of the Al-ZnO nanoparticles using the plant extracts of *Cucumis maderaspatanus*.

S. No	Name of the Organisms	Zone of Inhibition (mm-cm)		Positive Control
		5 mg/mL	10 mg/mL	
1.	*K. bacillus*	1.05 ± 0.02	2.65 ± 0.15	4.60 ± 0.07
2.	*E. coli*	NA	NA	2.10 ± 0.01
3.	*V. cholera*	1.10 ± 0.01	1.98 ± 0.03	NA
4.	*B. subtilis*	0.44 ± 0.08	1.01 ± 0.01	3.25 ± 0.01
5.	*S. aureus*	2.63 ± 0.02	4.01 ± 0.01	4.00 ± 0.02
6.	*S. mutans*	NA	0.67 ± 0.01	4.20 ± 0.34

Note: “NA” no inhibition zone.

**Table 2 nanomaterials-14-01851-t002:** Antifungal activity of Al-ZnO nanoparticles using the extracts of *Cucumis maderaspatanus*.

Name of the Organisms	Resistant Zone (mm-cm)		Positive Control
	5 mg/mL	10 mg/mL	
*Aspergillus flavus*	0.73 ± 0.04	1.25 ± 0.01	3.00 ± 0.00
*Candida albicans*	NA	NA	2.50 ± 0.61

Note: “NA” no inhibition zone.

**Table 3 nanomaterials-14-01851-t003:** Antioxidant activity results of Al-ZnO nanoparticles using the leaf extracts of *Cucumis maderaspatanus*.

S. No	Concentration (mg/mL)	Antioxidant Activity of % of Inhibition	Standard
1.	0.2	32.8	Ascorbic acid is used as a standard
2.	0.4	38.9	
3.	0.6	45.3	
4.	0.8	51.2	
5.	1.0	61.0	

## Data Availability

Data will be made available on request.

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
