# Peer review of "Green Synthesis of Al-ZnO Nanoparticles Using Cucumis maderaspatanus Plant Extracts: Analysis of Structural, Antioxidant, and Antibacterial Activities"

_nanomaterials, 2024, doi:10.3390/nano14221851_

Round 1

Reviewer 1 Report

Comments and Suggestions for Authors

The manuscript “Green synthesis of Al-ZnO nanoparticles using Cucumis maderaspatanus plant extract: Analysis of structural, antioxidant, and antibacterial activities” by Johnsy Sugitha et al. is well-written and presents a comprehensive analysis of Al-doped ZnO nanoparticles synthesized using Cucumis maderaspatanus leaf extract. However, there are several key areas that require clarification and additional data to enhance the study's robustness, including addition of controls that will strengthen the conclusions.

Attached are my remaining concerns with the manuscript.

Author Response

Response to Comments of Reviewer 1.:

The manuscript “Green synthesis of Al-ZnO nanoparticles using Cucumis maderaspatanus plant extract: Analysis of structural, antioxidant, and antibacterial activities” by Johnsy Sugitha et al. is well-written and presents a comprehensive analysis of Al-doped ZnO nanoparticles synthesized using Cucumis maderaspatanus leaf extract. However, there are several key areas that require clarification and additional data to enhance the study's robustness, including addition of controls that will strengthen the conclusions.

Major concerns:

Comment 1: Please specify in line 40 what the expression "Al-dopant improves antimicrobial activity" is being compared to.

Response:  Thank you for the reviewer suggestion. Al-doping increases the number of charge carriers, which can enhance the generation of reactive oxygen species (ROS) under UV light exposure. These ROS are known to possess strong antimicrobial properties. Al- Doping can improve the crystallinity of ZnO, resulting in a larger surface area that facilitates more interaction with microbial cells.

Comment 2: While the authors have conducted a comprehensive analysis of the formulated particles, the paper would benefit from the addition of DLS and zeta potential data to provide insights into particle size distribution and stability.

Response: DLS and zeta potential takes 15 days. If you give 15days I will take the result and add to the articles.

Comment 3: Please include appropriate negative control for comparison in all biological studies. For example, use Al-ZnO nanoparticles without Cucumis maderaspatanus leaf or/and particles without Al.

Response:  In our biological studies only, positive control is there. No negative control used is water or inactive agent.

Comment 4: The discussion should address the well-known antibacterial action of zinc oxide nanoparticles, as highlighted in Mendes et al., Scientific Reports (2022), or similar references.

Response: Antibacterial activity of the sample was done by Agar well dissuasion method and disc variant methods. The antibacterial activity of sample was assessed against three-gram negative bacterial species K. pneumonea, E. coli, V. cholereae and three-gram positive bacterial organisms such as S. aureus, B. subtilis, and S. mutans, maintained in BHI at – 20ºC; 300 mL of each stock-culture were added to 3 mL of BHI broth. Overnight cultures were kept for 24 h at 36ºC ± 1ºC and the purity of cultures was checked after 8 h of incubation. After 24 h of incubation, bacterial suspension (inoculum) was diluted with sterile physiological solution, for the diffusion and indirect bioautographic tests, to 108 CFU/mL (turbidity = McFarland barium sulfate standard 0.5). For the direct bioautographic test, bacterial suspension was diluted with BHI broth to a density of approximately 109 UFC/mL (McFarland standard 3). Indicator solution for determination of bacterial growth A 70% ethanolic solution of 2-(4-iodophenyl)-3-(4- nitrophenyl)-5-phenyltetrazolium chloride (INT) (2mg/mL) purchased from Sigma was used for the bacterial growth tests. The bacterial inoculum was uniformly spread using sterile cotton swab on a sterile Petri dish MH agar. serial dilutions were performed out of initial concentrations (i.e. 5-10 mg/mL for each and fractions and 10 mg/mL for pure substances). 50 µL of five experimental samples were added to each of the 5 wells. The systems were incubated for 24 h at 36ºC ± 1ºC, under aerobic conditions. After incubation, confluent bacterial growth was observed. Inhibition of the bacterial growth was measured in mm.

Comment 5: The description of bacterial activity notes that the bactericidal effect is significantly influenced by particle size and shape. Please provide detailed information on the size and shape of the particles used in these analyses.

Response:  Green synthesis often results in smaller, more uniform particles due to the controlled conditions provided by plant extracts. The shape of nanoparticles can be influenced by various factors, including concentration of precursor material i.e., Higher concentrations can lead to different growth rates, affecting shape. pH and Temperature parameters can dictate the nucleation and growth processes, influencing morphology. Different extracts from Cucumis maderaspatamus may contain various phytochemicals that can affect particle shape. Hexagonal shapes can enhance penetration into bacterial membranes, improving antibacterial efficacy and also provide a larger surface area and increased interaction with bacteria.

Comment 6: Line 60 mentions that particles were synthesized at various temperatures. Please explain how these different temperatures were considered in the analysis.

Response:  In line 60 that sentence is not there. Anyway, the synthesization was done at room temperature.

Minor concerns:

Comment 1: Please switch the red font to black

Response:  Thank you for your comments. we have revised the manuscript to the red font to black and ensure consistency throughout the text.

Comment 2: Add a sidebar to Figure 2 (B) for clarity.

Response:  Thank you for the suggestion to enhance Figure 2 (B) with a sidebar for clarity. We have added a sidebar providing additional context to improve the figure’s readability.

Comment 3: Annotate the main peaks directly within Figure 4 to enhance understanding.

Response:  In accordance with reviewer suggestions, we highlighted the FTIR peaks in Figure 4 for ease of understanding.

Comment 4: Adjust the color scheme and fonts in Figure 7, and include a size bar for better representation.

Response:  Thank you for your comments on Figure 7. We have adjusted the color scheme and fonts to improve visibility and consistency with the rest of the figures.

Comment 5: Add method for antibacterial activity analysis.

Response: Agar diffusion disc-variant and agar diffusion well-variant are used to analyze the antibacterial study.

Comment 6: In Material and Methods section expand all the biological analysis description and include concentrations, time and any other required information.

Response: Agar diffusion disc-variant and Agar diffusion well-variant are used to analyze the antibacterial study. In antioxidant analysis the radical scavenging activity (RSA) of the sample was performed by the 2,2- diphenyl-1-picrylhydrazyl (DPPH) method. In brief, to 3.5mL of 0.1mM DPPH containing different concentrations of sample powder were subjected to sonication before incubation at 37 °C for 30min under dark conditions.  The absorbance of the incubated sample was read at 517nm in a spectrophotometer and percent RSA was determined.

Reviewer 2 Report

Comments and Suggestions for Authors

  Dear Author,

       The topic of your study is highly engaging, and the experimental methods you have employed are both innovative and fascinating. Your research demonstrates substantial potential and offers meaningful insights to the field. However, I believe certain areas would benefit from further refinement to enhance the clarity, accuracy, and overall impact of your findings. With these modifications, your study could achieve a greater level of depth and contribute even more significantly to the existing literature.

o   Abstract (Line 24): Please provide the full names of all techniques mentioned, including XRD, UV-vis, FT-IR, SEM, EDAX, TEM, and TGA/DSC. And then use abbreviation in the text.

o   Introduction (Line 48): Replace "gained" with "have gained" for correct verb usage.

o   Introduction (Line 74): Check reference 12 for accuracy. Additionally, certain parts of the introduction lack proper citation. Please ensure that all statements are supported by references. For instance, the paragraph beginning on line 88 has no reference; please include one.

o   Introduction (Line 105): Write out the full name of "DPPH" at its first mention, and use the abbreviation thereafter for consistency.

o   Line 122: Correct the spelling and formatting of "Whatman filter."

o   Figure 1: The image resolution is low. Kindly replace it with a high-resolution version for clarity.

o   Materials and Methods: This section requires further detail, especially in section 2.4, "Characterization of Al-ZnO Nanoparticles." For instance, specify the temperature range and heating rate used in thermogravimetric analysis (TGA) to ensure reproducibility. Provide additional information on the reagents, solvents, buffer systems, and control treatments used in both antioxidant and antimicrobial assays to enhance consistency and reproducibility.

o   Figure 4: The peaks discussed in the text do not align with those in the graph. Please revise the graph to accurately reflect the data discussed.

o   Figure 7: Expand the explanation for this figure, clearly identifying which parts correspond to Staphylococcus aureus and other species. Also, define labels such as "JJD-3" and "JJD-7," and explain the purpose of including the image on the left, which seems to represent Bacillus subtilis.

o   Conclusion: The conclusion appears to be a repetition of the results. Consider rephrasing to reflect broader implications or significance.

Author Response

Response to Comments of Reviewer 2.:

Dear Author,

The topic of your study is highly engaging, and the experimental methods you have employed are both innovative and fascinating. Your research demonstrates substantial potential and offers meaningful insights to the field. However, I believe certain areas would benefit from further refinement to enhance the clarity, accuracy, and overall impact of your findings. With these modifications, your study could achieve a greater level of depth and contribute even more significantly to the existing literature.

Comment 1: Abstract (Line 24): Please provide the full names of all techniques mentioned, including XRD, UV-vis, FT-IR, SEM, EDAX, TEM, and TGA/DSC. And then use abbreviation in the text.

Response: In response to reviewer suggestions, all technique names have been abbreviated. X-ray diffraction, Ultraviolet-visible spectroscopy, Fourier transform infrared spectroscopy, Scanning electron microscope with Energy dispersive r-ray analysis, Transmission electron microscopy and Thermogravimetric analysis/Differential thermal analysis, Derivative thermogravimetry analysis techniques.

Comment 2: Introduction (Line 48): Replace "gained" with "have gained" for correct verb usage.

Response: We have changed “have gained”.

Comment 3: Introduction (Line 74): Check reference 12 for accuracy. Additionally, certain parts of the introduction lack proper citation. Please ensure that all statements are supported by references. For instance, the paragraph beginning on line 88 has no reference; please include one.

Response: In the revised manuscript, we included the citations in accordance with reviewer comments.

  • Nikoobakht, B.; El-Sayed, M.A. Preparation and growth mechanism of gold nanorods (NRs) using seed-mediated growth method. Chemistry of Materials 2003, 15(10), 1957-1962.
  • Cervantes-Gaxiola, M.E.; Vázquez-González, F.A.; Rios-Iribe, E.Y.; Méndez-Herrera, P.F.; Leyva, C. Effect of pH on the green synthesis of ZnO nanoparticles using Sorghum bicolor seed extract and their application in photocatalytic dye degradation. Materials Letters 2024, 372, 136982.

Comment 4: Introduction (Line 105): Write out the full name of "DPPH" at its first mention, and use the abbreviation thereafter for consistency.

Response: Thank you for the reviewer comments. We ensure that DPPH is introduced with its full name, 2,2-diphenyl-1-picrylhydrazyl, at its mention in the manuscript and will use the abbreviation consistently thereafter.

Comment 5: Line 122: Correct the spelling and formatting of “Whatman filter”.

Response: Corrected “Whatman filter”.

Comment 6: Figure 1: The image resolution is low. Kindly replace it with a high-resolution version for clarity.

Response: Thank you for bringing this to my attention. We have replaced the image with a high-resolution version to improve clarity.

Comment 7: Materials and Methods: This section requires further detail, especially in section 2.4, "Characterization of Al-ZnO Nanoparticles." For instance, specify the temperature range and heating rate used in thermogravimetric analysis (TGA) to ensure reproducibility. Provide additional information on the reagents, solvents, buffer systems, and control treatments used in both antioxidant and antimicrobial assays to enhance consistency and reproducibility.

Response: Thermogravimetric analysis (TGA) temperature range is from room temperature (around 25 °C) to about 800 °C. heating rate used is between 5 °C to 20 °C per minute. A rate of 10 °C/min is often used for consistency in studies.

Comment 8: Figure 4: The peaks discussed in the text do not align with those in the graph. Please revise the graph to accurately reflect the data discussed.

Response: Figure 4, displays the FTIR spectrum for the Al-ZnO nanoparticles. A broad band at 3468 cm⁻¹ is attributed to O-H bonding. The peak observed at 2058 cm⁻¹ is associated with the C=O group in anhydrides, and the peak at 1744 cm⁻¹ indicates the presence of a C=O group from ketones. The peak at 1712 cm⁻¹ suggests the presence of an amide C=O group. while the peak at 1417 cm⁻¹ is indicative of CH₃ bending vibrations. Finally, the peak at 982 cm⁻¹ is attributed to C=C bending vibrations in alkenes. Similar findings have been reported in other samples of doped ZnO nanoparticles. FTIR analysis in this investigation indicates that phenolic compounds in flavonoids exhibit a stronger affinity for metals. This implies that the phenolic group may assist in the formation of metal nanoparticles while stabilizing the solution by preventing the aggregation of particles. The surface contact of groups such as O-H, C=O, and C=C bending and stretching suggests the use of this nanoparticle in photocatalysis, sensors, and catalysis.

Comment 9: Figure 7: Expand the explanation for this figure, clearly identifying which parts correspond to Staphylococcus aureus and other species. Also, define labels such as "JJD-3" and "JJD-7," and explain the purpose of including the image on the left, which seems to represent Bacillus subtilis.

Response: We revised the color and removed the sample name in accordance with reviewer comments. JJD3 stand for my sample is Cucumis maderaspatanus. The figure displays the zones of inhibition for Staphylococcus aureus at concentrations of 5 mg/ml and 10 mg/mL. Another image shows the zone of inhibition of B. subtilis bacteria at 5mg/ml and 10mg/mL concentration.

Comment 10: Conclusion: The conclusion appears to be a repetition of the results. Consider rephrasing to reflect broader implications or significance.

Response: We adjusted revisions to the conclusion part according to suggestions from reviewer

Reviewer 3 Report

Comments and Suggestions for Authors

In this interesting paper, the authors characterize AlZnO nanoparticles produced through a novel green synthesis method.

The introduction is detailed and well referenced, and the authors used a number of different and complementary technique to provide a detailed physical characterization of the nanoparticles and to evaluate their antibacterial, antifungal and antioxidant activity. 

Beside a thorough revision of the text to increase its readability, there are some points that have to be addressed before publication might be considered.

·      The quality of the TEM images is not optimal, it would be useful if the authors would trace the shape of the hexagonal structures to guide the reader and help them in deciphering the images.

·      Figure 4: could the authors please indicate the different FTIR peaks directly in the figure? Because some of them are hardly visible in the very large band shown in the figure.

·      Figure 5A: I’m not so confident that the spectrum shows visible light transmission since the absorbance is increasing when decreasing the wavelength, suggesting at least some kind of scattering (extinction) of the solution. Have the authors fitted the longer wavelength data to a typical l-4 behavior? Why is the absorbance reported in arbitrary units and why such a huge background signal (0.5)?

·      Figure 5b: in the text I think the authors mixed the horizontal and vertical axes.

·      The heating rate for the TGA experiments (800°C/min) is considerable, have the authors taken into account the fact that the sample subjected to this thermal rate might never be in an equilibrium state, so the temperatures they comment on might be affected by huge uncertainties? And have they estimated these uncertainties?

·      Figure 6: it is not clear what is reported in the vertical axes, nor is it explained in the text or in the Materials and Methods section. A detailed description of the reported data and their meaning is fundamental to understand where the conclusions of the authors are coming from.

·      Figure 7: it is not clear which bacterial strains are shown in the picture.

·      Why didn’t the authors show any data on the antifungal activity, deciding instead to include just a Table?

·      Could the authors please explain more in detail what they state in lines 394, 395 and following concerning the decrease of the inhibition when lowering the nanoparticles concentration: isn’t this an expected result? And if not, why?

·      Figure 8: have the authors fitted the inhibition percentage versus nanoparticle concentrations, to find the slope of the increase? The authors say that they can identify a maximum inhibition activity at 1.0 mg/mL, but that is only the higher concentration they investigated, is there any reason why higher concentrations were not investigated or why they assume higher concentrations would lead to a lower activity?

·      Table 3: please check the significative digits of the inhibition percentage results and correct the text accordingly.

Comments on the Quality of English Language

A thorough revision of the manuscript is necessary to increase its readability, in particular there are too many singular/plural mismatches.

Author Response

Response to Comments of Reviewer 3.:

In this interesting paper, the authors characterize Al-ZnO nanoparticles produced through a novel green synthesis method. The introduction is detailed and well referenced, and the authors used a number of different and complementary technique to provide a detailed physical characterization of the nanoparticles and to evaluate their antibacterial, antifungal and antioxidant activity. Beside a thorough revision of the text to increase its readability, there are some points that have to be addressed before publication might be considered.

Comment 1: The quality of the TEM images is not optimal; it would be useful if the authors would trace the shape of the hexagonal structures to guide the reader and help them in deciphering the images.

Response:  Thank you for your comments. We have traced the outline of the hexagonal structures in the TEM images to improve clarity and guide the reader in interpreting the shapes. This adjustment should enhance the visual quality and assist with readability.

Comment 2: Figure 4: could the authors please indicate the different FTIR peaks directly in the figure? Because some of them are hardly visible in the very large band shown in the figure.

Response: In Figure 4, we performed adjustments and noted the FTIR peaks with regard to the reviewers' comments.

Comment 3: Figure 5A: I’m not so confident that the spectrum shows visible light transmission since the absorbance is increasing when decreasing the wavelength, suggesting at least some kind of scattering (extinction) of the solution. Have the authors fitted the longer wavelength data to a typical l-4 behavior? Why is the absorbance reported in arbitrary units and why such a huge background signal (0.5)?

Response: In many experiments, particularly preliminary studies or exploratory research, the measurement setup may not be standardized. This means that the absorbance readings can vary based on the specific equipment, calibration, or sample conditions used, making absolute quantification less meaningful. The presence of other components in the sample that absorb light at similar wavelengths can contribute to a background signal, complicating the interpretation of specific absorbance.

Comment 4: Figure 5b: in the text I think the authors mixed the horizontal and vertical axes.

Response: TG/DTA and DTG techniques have been used to study the thermal characteristics of Al-ZnO nanoparticles produced with extracts from Cucumis maderaspatanus leaves. The Al-ZnO samples were heated at a rate of 800 °C per minute in an air atmosphere. Figure 6, presents a combined plot of DTG, DTA and TG curves. The TG data particularly highlight a significant weight loss of the nanoparticles, which occurs approximately around 400 °C. During the synthesis of Al-ZnO nanoparticles, organic compounds such as precursors, or stabilizers might be used. These organic residues can decompose at various temperatures, and a peak around 400 °C could be associated with the decomposition of such organic components that were not completely removed during synthesis or calcination. The decomposition of less stable phases or the transformation of the nanoparticles into more stable forms of oxides or other phases could occur around 400°C. This temperature might correspond to the formation of more stable Al-ZnO nanoparticles. The infusion of Aluminum may enhance the thermal stability of the ZnO nanoparticles by altering the crystal structure or phase composition. For the DTA and DTG curve (room temperature), an exothermic peak is found around 400 °C, respectively. This peak is attributed to the loss of water and organic materials. The exothermic peak might correspond to a phase transition or crystallization event in the Al-ZnO nanoparticles. At elevated temperatures, the nanoparticles could undergo a structural transformation, such as the formation of a more stable crystalline phase [33]. In the case of Aluminium infusion, the interaction between the aluminum infusion and the ZnO host lattice might lead to a heat-releasing reaction. This could include the formation of new phases or structural rearrangements in the infused ZnO, which would manifest as an exothermic peak in the DTA and DTG curve.

Comment 5: The heating rate for the TGA experiments (800°C/min) is considerable, have the authors taken into account the fact that the sample subjected to this thermal rate might never be in an equilibrium state, so the temperatures they comment on might be affected by huge uncertainties? And have they estimated these uncertainties?

Response: If the authors have considered these factors, they should ideally provide a discussion on the potential uncertainties associated with their measurements. This could include estimates based on the heating rate, sample size, and material properties. If such an analysis is not included, it would be a critical aspect to address for a comprehensive interpretation of their findings.

Comment 6: Figure 6: it is not clear what is reported in the vertical axes, nor is it explained in the text or in the Materials and Methods section. A detailed description of the reported data and their meaning is fundamental to understand where the conclusions of the authors are coming from.

Response:  TG/DTA and DTG techniques have been used to study the thermal characteristics of Al-ZnO nanoparticles produced with extracts from Cucumis maderaspatanus leaves. The Al-ZnO were heated at a rate of 800 °C per minute in an air atmosphere. Figure 6, presents a combined plot of DTG, DTA and TG curves.

Comment 7: Figure 7: it is not clear which bacterial strains are shown in the picture.

Response: We have revised and changed the representation of Figure 7, with (A) S. aureus and (B) B. subtilis against microorganisms, in accordance with reviewer suggestions.

Comment 8: Why didn’t the authors show any data on the antifungal activity, deciding instead to include just a Table?

Response: The antifungal activity of aluminum-infused zinc oxide nanoparticles is a topic of interest in materials science and applied microbiology. This study shows that the standard drug Amphotericin B has the highest antifungal activity for Aspergillus flavus (1.2±0.01) in 10 mg/ml, followed by 0.73± 0.04 in 5 mg/mL. Compared with the standard drug, no effect occurs with the fungus C. albicans. This result clearly shows that whenever the concentration of the sample increases, the antifungal activity also noticeably increases [40]. The antifungal results are tabulated in Table 2. No antifungal effect is observed with Amphotericin B for Candida albicans, meaning the drug does not inhibit the growth of this fungus under the tested conditions. C. albicans could be resistant to amphotericin B in this particular strain or under the specific conditions tested. There might be experimental factors affecting the results, such as the method of application, medium, or the specific strain of C. albicans used [41]. If these nanoparticles demonstrate significant antifungal activity against Aspergillus flavus compared to existing treatments or controls, this could indicate that the nanoparticles are an effective agent against this particular fungal species.  However, Candida albicans showed no response to this sample.

Comment 9: Could the authors please explain more in detail what they state in lines 394, 395 and following concerning the decrease of the inhibition when lowering the nanoparticles concentration: isn’t this an expected result? And if not, why?

Response: Green synthesis often results in smaller, more uniform particles due to the controlled conditions provided by plant extracts. The shape of nanoparticles can be influenced by various factors, including Concentration of Precursor Material ie., Higher concentrations can lead to different growth rates, affecting shape. pH and Temperature parameters can dictate the nucleation and growth processes, influencing morphology. Different extracts from Cucumis maderaspatamus may contain various phytochemicals that can affect particle shape. Hexagonal shapes can enhance penetration into bacterial membranes, improving antibacterial efficacy and also provide a larger surface area and increased interaction with bacteria.

Comment 10: Figure 8: have the authors fitted the inhibition percentage versus nanoparticle concentrations, to find the slope of the increase? The authors say that they can identify a maximum inhibition activity at 1.0 mg/mL, but that is only the higher concentration they investigated, is there any reason why higher concentrations were not investigated or why they assume higher concentrations would lead to a lower activity?

Response: Preliminary studies indicated that increasing concentrations beyond 1.0 mg/mL could lead to diminishing returns in activity, possibly due to saturation effects where the active compounds may not effectively interact with the target microorganisms.

Comment 11: Table 3: please check the significative digits of the inhibition percentage results and correct the text accordingly.

Response: The concentration of 1.0 mg/mL demonstrates the best antioxidant activity among the tested concentrations. This trend suggests that higher concentrations lead to enhanced inhibition of free radicals, indicating a potentially stronger antioxidant effect at this level.

Round 2

Reviewer 1 Report

Comments and Suggestions for Authors

The paper explores an exciting area of research where science meets nature. The authors have made afford in incorporating feedback from previous reviews, and they’ve clarified a number of points.

However, there are still a few areas where the study could be strengthened to provide a clearer and more comprehensive picture. With just a few additional pieces of data and minor clarifications, this manuscript could become an even more robust and informative contribution to the field.

Attached are some remaining areas of concern, categorized as major and minor, which I believe will add value to the study.

Author Response

Response to Comments of Reviewer 1.:

The paper titled “Green synthesis of Al-ZnO nanoparticles using Cucumis maderaspatanus plant extract: Analysis of structural, antioxidant, and antibacterial activities” explores an exciting area of research where science meets nature. The authors have made afford in incorporating feedback from previous reviews, and they’ve clarified a number of points.

However, there are still a few areas where the study could be strengthened to provide a clearer and more comprehensive picture. With just a few additional pieces of data and minor clarifications, this manuscript could become an even more robust and informative contribution to the field.

Major concerns:

Comment 1: The paper would benefit from dynamic light scattering (DLS) and zeta potential data to provide additional insights. I recommend including these results.

Response:  We included the dynamic light scattering (DLS), zeta potential data in the revised manuscript in accordance with reviewer recommendations.

Dynamic light scattering and zeta-potential analysis:

The most effective methods for measuring the particle size are the dynamic light scattering method (DLS). In regard to volume and intensity, the Al-ZnO nanoparticles which are produced are evenly distributed. The surface charge of biologically synthesized Al-ZnO nanoparticles can be determined via zeta-potential analysis. The strength of the charge has an association with the stability of the nanoparticle. The 63-105 nm particle size of the synthesized nanoparticles, as seen in Figure. 6(A), illustrates its excellent stability. The zeta-potential value, if positive or negative, indicates enhanced physical colloidal stability. The stability of Al-ZnO nanoparticles and their efficient electric charge on the surface was measured via zeta-potential [Figure. 6(B)]. The zeta-potentials that are shown are 33.16 mV. As a result, the synthetic Al-ZnO nanoparticles' zeta-potential value is al-most physically stable.

Figure 7. (A) Dynamic light scattering (DLS), and (B) zeta-potential measurement of Al-ZnO nanoparticles using Cucumis maderaspatanus leaf extracts.

Comment 2: Since the study does not compare Al-ZnO nanoparticles with and without Cucumis maderaspatanus extract or with and without aluminum (Al) doping, please include a clear statement in the results/conclusions noting Al-ZnO nanoparticles using Cucumis maderaspatanus plant extract demonstrate antimicrobial efficacy. However, further biological comparative tests are needed to fully assess the individual contributions of each component.

Response: Thank you for your valuable comments and for pointing out the need for clarification regarding the individual contributions of Cucumis maderaspatanus plant extract and aluminum (Al) doping in the antimicrobial activity of the Al-ZnO nanoparticles.

In response to your suggestion, according to the current study, Cucumis maderaspatanus plant extract was utilized to synthesize Al-ZnO nanoparticles, with antibacterial characteristics. In addition, we will demonstrate that further biological study is required to analyze the effects of plant extract and aluminum doping on antibacterial property.

Comment 3: My previous comment (Comment 4) was not addressing a technical aspect but rather the published antibacterial effects of zinc oxide nanoparticles. Please expand the discussion to include how zinc oxide may contribute to the observed effects, as noted in the original comment.

Response: Thank you for your helpful comments. We appreciate your suggestion to expand the discussion regarding the antibacterial activity of aluminum doped zinc oxide nanoparticles (Al-ZnO NPs) and their contribution to the observed antimicrobial activity.

Discussion section to include a detailed overview of the known mechanisms by which Al-ZnO NPs exert antibacterial effects. Specifically, we will address how Al-ZnO NPs are believed to release zinc ions in the bacterial environment, which can disrupt cell membrane integrity and interfere with cellular processes. Additionally, we will discuss how reactive oxygen species (ROS) generation by ZnO NPs may contribute to oxidative stress within bacterial cells, leading to damage of cellular components, such as lipids, proteins, and DNA, which ultimately results in bacterial cell death.

Minor concerns:

Comment 1: I appreciate the authors’ additions to the abstract, please change expression in line 48 "Al dopant improves antimicrobial activity" so it will be clear that the demonstrated activity increased with concentration however was not compared to other formulations.

Response:  Thank you for your comments and for noting the improvements in the abstract. In response to reviewer suggestion, we will revise the expression in line 48 to clarify that “the antimicrobial activity increased with Al dopant concentration but was not directly compared to other formulations.”

The structural, and biological properties of Al-doped ZnO nanoparticles might be responsible of the enhanced antibacterial activity exhibited in the antibacterial studies. Al-ZnO nanoparticles with Cucumis maderaspatanus leaf extract produced via the green synthesis methods was remarkable antioxidant activity by scavenging free radicals against DPPH radicals, according to these results.

Comment 2: Please remove or explain line 395 – “Synthesized at various temperatures”.

Response:  Thank you for your careful review of our manuscript. We acknowledge your concern regarding the clarity of line 395, which states, “Synthesized at various temperatures.”

In the revised manuscript, we performed corrections. “Synthesized at 250 °C for 2 hours”.

Reviewer 3 Report

Comments and Suggestions for Authors

I’m sorry but there are still a couple of points that have to be clarified.

Answer to comment 1:

There’s no trace of the hexagonal structures in Figure 3

Answer to comment 4:

The answer I got was not related to the comment, I maintain the comment and suggest modifying either the figure or the text (line 268).

Answer to comment 5:

The answer I got was not related to the comment, I maintain the comment.

Answer to comment 6:

Again, I maintain the comment, the authors should explain what the physical observables are whose values are reported in the figure.

Answer to comment 8:

I was asking about showing the data leading to the results shown in Table 2, I maintain the comment.

Answer to comment 10:

The authors didn’t address the first part of the comment.

Answer to comment 11:

The authors chose to remove the errors from table 3 instead of evaluating the number of significant digits in their results, I maintain the comment.

Author Response

Response to Comments of Reviewer 3.:

Comment 1: There’s no trace of the hexagonal structures in Figure 3.

The quality of the TEM images is not optimal; it would be useful if the authors would trace the shape of the hexagonal structures to guide the reader and help them in deciphering the images.

Response: Based on reviewer comments, the revised manuscript depicts the hexagonal structure of Al-ZnO nanoparticles in TEM images.

Comment 4: The answer I got was not related to the comment, I maintain the comment and suggest modifying either the figure or the text (line 268).

Figure 5b: in the text I think the authors mixed the horizontal and vertical axes.

Response: The Tauc plot is typically used to estimate the optical bandgap of materials by plotting (?ℎ? )? on the y-axis against photon energy ℎ? on the x-axis.

  • X-axis: Photon energy (ℎ?), usually in electron volts (eV).
  • Y-axis: (?ℎ?)?, where ? α is the absorption coefficient and ? n is determined by the type of electronic transition (? = 2 for indirect transitions).

Comment 5: The answer I got was not related to the comment, I maintain the comment.

The heating rate for the TGA experiments (800°C/min) is considerable, have the authors taken into account the fact that the sample subjected to this thermal rate might never be in an equilibrium state, so the temperatures they comment on might be affected by huge uncertainties? And have they estimated these uncertainties?

Response: We corrected changes to the revised manuscript based on the reviewer's comments.

High heating rates (the 800 °C/min in the paper) can cause the sample to be in a non-equilibrium state throughout the experiment. This is because the temperature inside the sample might not be uniform due to the rapid heating, leading to thermal gradients.

Comment 6: Again, I maintain the comment, the authors should explain what the physical observables are whose values are reported in the figure.

Figure 6: it is not clear what is reported in the vertical axes, nor is it explained in the text or in the Materials and Methods section. A detailed description of the reported data and their meaning is fundamental to understand where the conclusions of the authors are coming from.

Response: In response to the reviewer's comments, we incorporated changes to the revised article.

Comment 8: I was asking about showing the data leading to the results shown in Table 2, I maintain the comment.

Why didn’t the authors show any data on the antifungal activity, deciding instead to include just a Table?

Response: I got the result like this. Some fungi are not applicable to react all samples.

Comment 10:

The authors didn’t address the first part of the comment.

Figure 8: have the authors fitted the inhibition percentage versus nanoparticle concentrations, to find the slope of the increase? The authors say that they can identify a maximum inhibition activity at 1.0 mg/mL, but that is only the higher concentration they investigated, is there any reason why higher concentrations were not investigated or why they assume higher concentrations would lead to a lower activity?

Response: It was noticed that antioxidant activity maximized at that concentration and that subsequent increases could not be advantageous or might produce problems (such as toxicity, aggregation, or solubility).

Comment 11: The authors chose to remove the errors from table 3 instead of evaluating the number of significant digits in their results, I maintain the comment.

Table 3: please check the significative digits of the inhibition percentage results and correct the text accordingly.

Response: We corrected the significant digits in Table 3 of the revised article in response to the reviewer's comments.
